# A Mathematical Model for Sublimation of a Thin Film in Trace Explosive Detection Problem

**DOI:** 10.3390/molecules27227939

**Published:** 2022-11-16

**Authors:** Olga B. Kudryashova, Sergey S. Titov

**Affiliations:** 1Institute for Problems of Chemical and Energetic Technologies, Siberian Branch of the Russian Academy of Sciences (IPCET SB RAS), St. Socialist, 1, 659322 Biysk, Russia; 2Physics and Technology Faculty, National Research Tomsk State University, Lenin Avenue 36, 634050 Tomsk, Russia

**Keywords:** sublimation, explosives, thin film, detection techniques

## Abstract

Here, we introduce an advanced mathematical model for the sublimation of thin films of explosives. The model relies on the Hertz–Knudsen–Langmuir (HKL) equation that describes the vaporization rate of an explosive and controls the mass exchange between the surface and the ambient air. The latest experimental data on sublimation and diffusion of 2,4,6-trinitrotoluene (TNT) monocrystals were factored in, as well as the data on the sublimation rate of hexogen (RDX), octogen (HMX), and picramide (TNA) traces. To advance the mathematical model we suggested previously, we took into account the structure of a substrate on which a thin explosive layer was deposited. The measurement problem of the sublimation rate and limits of an explosive arises from developing and advancing remote detection methods for explosives traces. Using mathematical modelling, we can identify a detectable quantity of a specific explosive under given conditions. We calculated the mass of the explosive in the air upon sublimation of thin explosive films from the surfaces over a wide range of the parameters in question and made conclusions regarding the application limits of the devised standoff trace explosive detection techniques.

## 1. Introduction

Because of the need to support defense and security, an important challenge is to detect trace explosives, including those being low in volatiles. When organizing antiterrorist controls, different methods are employed to record explosives molecules in the air upon sublimation of traces of such materials from the surface of objects [1,2,3,4]. The efficiency of trace explosive detection methods is influenced by the sublimation kinetics of material taken up from the surface of objects.

At normal temperatures, the pressure of trace vapors of most common explosives is small, hindering their detection [1,5]. For instance, the pressure of saturated vapors at 25 °C is ~1.7 × 10^−3^ Pa for TNT (trinitrotoluene), while that for RDX (hexogen) is about 4 × 10^−6^ Pa [1,6]. The vapor detection methods are improving and their threshold sensitivity is rising. Standoff explosive vapor detectors that are based on the LIDAR principle, as an example, can detect TNT vapors at a concentration below 10–12 g/cm^3^ [7,8,9]. However, one should know whether explosive traces can be detected under specified conditions; whether the resulting vapors will be enough for detection; and at which instant of time, once an explosive trace has originated, the vapors will reach a sufficient concentration in the air.

The explosive trace is the thinnest film of an explosive on the surface. For instance, it was reported [10] that the quantity of an explosive left behind by a single fingerprint on the surface of an item is ~10 μg. The vaporization processes of thin films of explosives are distinct from those of great volumes, and this issue has been underexplored in the literature. The phase transition temperature of the thin film is much lower than that in a large volume of an explosive [11]. Explosives may differ in phase state under normal conditions. The interaction of thin molecular layers with the surface is more intense than that in thick layers, while vaporization/sublimation commences already at room temperature.

The sublimation rate of an explosive, in particular TNT and RDX, was examined in several studies [12,13,14,15,16,17,18,19,20]. The sublimation rate of explosives with relatively thick films (considerably thicker than 3 nm) on different substrates was considered in [13,14,15]. In these instances, the sublimation of molecules occurs from the surface of an explosive layer, and the substrate molecules exert an impact only at the final stage of the process. Pacheco-Londoño et al. [16] not only experimentally obtained sublimation energies of explosives but also detected an impact of the substrate (steel plate) on the desorption kinetics of a thin RDX layer. That study also comprehensively reviewed experimental findings on the sublimation heat of explosives, which were obtained by different authors.

The results obtained by different authors differ. These differences can most often be explained by different experimental methods used in different studies. In addition, those can be explained by the probable effect of the substrate from which sublimation took place. Pacheco-Londoño et al. [16] derived experimental data on the sublimation heat of a few explosives by the GAP method (Fiber Optic Coupled–Grazing Angle Probe). The desorption heat of RDX from a metal substrate was also measured. In the case of the RDX layer, sublimation was shown to occur on the metal surface first. Then, the interaction effect between the explosive and the substrate became visible and desorption occurred. RDX might be bound to the metal surface via the NO2 group. The Arrhenius law was employed for the modeling of the desorption reaction kinetics. The obtained energies of RDX desorption and sublimation from the steel surface in that study appeared to be similar. This corroborates the importance of factoring in the interaction between an explosive and a surface when looking into the sublimation process of thin explosive layers.

In the previous study [17], we proposed a mathematical model for the vaporization of a thin explosive film and revealed basic parameters influencing that process: ambient temperature, phase transition heat, and so on. In doing so, we took into account the impact of explosive vapors in the ambient air. We demonstrated that the vapors of trace explosives turned out to be insufficient for detection in most cases. The modelling found a parameter responsible for the interaction between the substrate surface and the explosive layer molecules. That parameter was a constant in the proposed model, and this constant can be measured experimentally for various combinations of a substrate and an explosive. To further advance the mathematical model, we suggest that this parameter be examined so that the substrate surface structure is factored in.

The present study aimed to mathematically model the sublimation of a thin film of an explosive, with the surface structure taken into account. Based on the mathematical model, we sought to identify conditions under which explosive vapors would be sufficient for detection when explosive traces vaporized.

## 2. Problem Setting

The mathematical model for sublimation is a continuation of the model we reported previously [17] and controls for the surface structure.

### 2.1. Assumptions

The model is underpinned by the mass and energy conservation laws.

A thin film is considered to be an explosive layer that is below 1 μm in thickness and deposited onto a substrate surface. It is important to note that the characteristic size of the film with area S significantly exceeds the film thickness *h*, $\sqrt{S}>>h$.

As experimentally shown by atomic force microscopy in the recent study [18], a decrease in the layer dimensions is caused by a reduction in the layer surface area rather than the layer thickness. The layer gradually “dries out” across the edges. Therefore, let us consider that surface area *S* decreases from the initial value of *S*_0_ during the sublimation and hence mass *m* decreases from the initial value of *m*_0_, while thickness *h* remains unchanged.

Consider that an explosive film with thickness h and surface area *S*_0_ has originated on a surface in the ambient air at rest at the initial instant. There are no explosive vapors in the air at the initial instant. The volume of the ambient air is unlimited (much greater than that of the diffusion boundary layer). The air temperature is *T*_0_.

The important characteristic of the surface onto which an explosive has been deposited is its porosity/roughness. The molecules are less tightly bound to an ideally smooth surface than with a surface, being meso- and microporous. The micropores/capillaries are easily filled with an explosive when contacted with the surface if the substance wets the surface. An adsorption (physicochemical) bond is formed that helps the explosive be retained in the pores. Such pores require greater energy than the smooth surface for the removal of the explosive. Therefore, the sublimation kinetics is contingent on the surface porosity. Let the porosity be assumed to be the specific surface area of the substrate, *S_ssa_*. The relation of the explosive layer surface area with the substrate surface area, *S*_0_/*S_s_*, is more informative. It is this parameter that will be used in the considerations below.

The film undergoes sublimation, whereby a molecular layer of explosive vapors (diffusion boundary layer) is generated over the film surface. In that layer, the vapor concentration diminishes from the maximum near the surface to the minimum in the ambient. There are no explosive molecules in the ambient at the instant time. They appear when the film is vaporizing, and if the volume is confined, their presence has to be taken into account. The molecules produce a vapor pressure that slows down the sublimation process. At some point, there may come an equilibrium state where the sublimation rate equals the condensation rate and the process halts. If the volume is not confined, the vapor molecules escape freely. Then the vapor pressure in the ambient is negligible. The sublimation rate exceeds the condensation rate and the film will be vaporized completely. However, the rate of this process can be so small that the film will disappear completely only in the perspective of infinite time.

It is necessary to find the sublimation rate and the vapor mass as a function of an explosive’s characteristics, ambient temperature, and substrate specific surface area. In addition, an inverse problem has to be settled, i.e., to identify the conditions under which this detection is feasible if the vapor mass detectable by the instrument is known.

### 2.2. Basic Equations

In line with the Hertz–Knudsen–Langmuir equation for the mass rate of vaporization:(1)$$v=-\frac{dm}{dt}=\alpha S\left(p_{s}-p_{e}\right)\sqrt{\frac{M}{2\pi RT}},$$
where *M* is the molecular weight of the explosive, *p_s_* is the partial pressure of saturated vapor in the diffusion boundary layer, *p_e_* is the explosive vapors in the ambient air, *R* is the universal gas constant, *T* is the temperature, *m* = ρ·*S*·*h* is the mass of the explosive, ρ is the density of the explosive, *S* and *h* is the area and the thickness of the film layer. The mass of the explosive will be changed by the reduction in the surface area, while the layer thickness will remain unchanged (Section 2.1).

Coefficient α is the portion of the flow of molecules turning into the vapor. Correspondingly, 1 − α is the portion of molecules condensing on the surface. This coefficient characterizes, first of all, the association between the explosive molecules and the surface. As noted above, the higher the substrate specific surface area *S_s_* with respect to the film surface area, the better the explosive molecules retained therein and, conversely, the fewer the explosive molecules escape from the surface. Therefore, it can be hypothesized that in regard to the portion of molecules condensing on the surface, 1 − α, is proportional to *S_s_*/*S*_0_; from here, α = 1 − α_0_·*S_s_*/*S*_0_. For an ideally smooth surface, having no pores, cracks, or roughness, *S_s_* = *S*_0_ and the explosive film will be least absorbed by the substrate. In practice, this idealized case is always *S_s_* > *S*_0_.

Equation (1) indicates that the direction of the flows and the amount of the explosive carried over from and to the surface is defined by the pressure difference of the explosive vapors directly above the surface in the diffusion boundary layer, *p_s_*, and beyond that layer in the ambient air, *p_e_*. Since the explosive vapors are initially absent in the ambient air and the sublimation proceeds into the open space, whose volume is much greater than that of the diffusion boundary layer, this term can be ignored in the equation.

Then, Equation (1) can be written with respect to the change in the surface area populated with the explosive:(2)$$\frac{dS}{dt}=-\alpha \frac{S\cdot p_{s}}{h\cdot \rho }\sqrt{\frac{M}{2\pi RT}},$$

In line with the Clausius–Clapeyron equation, we shall determine the partial pressure of the saturated vapor:(3)$$p_{s}=A\cdot \exp\left(-\frac{MH}{RT}\right),$$
where *H* is the heat of sublimation [J/kg] and *A* is the constant [Pa].

The analytical solution to Equation (2) has the form of:(4)$$S=S_{0}e^{-\beta t},$$
where $\beta =\alpha \frac{p_{s}}{h\cdot \rho }\sqrt{\frac{M}{2\pi RT}}$, [1/s]. The higher the value of β, the faster the decline in the explosive layer surface area. Note that Equation (4) is also true for the mass of the explosive, and the following can be written if we multiply the left-hand and right-hand sides by h·ρ:(5)$$m=m_{0}e^{-\beta t}.$$

Equation (5) can be written for the relative change in mass Δ*m*/*m*_0_:(6)$$\frac{\Delta m}{m_{0}}=e^{-\beta t}-1$$.

It follows from Equation (5) that the film will be vaporized completely (*m* = 0) only if *t* → ∞. The quantity *t_e_* = 1/β shows how long it will take for the film mass to decrease by the *e* times, which can be considered a typical time of sublimation. That said, the film mass will become equal to *m_e_* = *m*_0_/*e*, while the vapor mass will be *m_pe_* = *m*_0_ − *m_e_* = *m*_0_ (1 − 1/*e*) = 0.6321·*m*_0_.

At β → 0, the explosive mass does not change from the initial one. Since β~α, parameter β also becomes small at small values of parameter α. As noted herein above, there are grounds to assume that for the portion of molecules condensing on the surface, 1 − α, is proportional to *S_s_*/*S*_0_, where *S_s_* is the substrate surface area (inclusive of the surface of micropores, cracks, and roughness). Therefore, the greater the *S_s_*, the slower the sublimation process, through to its practical absence. This is due to the explosive molecules retaining well on the vast surface of the substrate. Parameter β also relies on temperature. The parameter is inversely proportional to the root of absolute temperature, while the saturated vapor pressure *p_s_* decreases with the temperature according to the exponential law (3). This mathematical relationship reflects the physical fact that the sublimation is slower at lower temperatures.

If the quantity of vapors sufficient for detection by any method is known, one can estimate the time within which the specified value of Δ*m_d_* is achieved. From Equation (6) is derived:(7)$$t_{d}=-\frac{1}{\beta }\ln\left[1-\frac{\Delta m_{d}}{m_{0}}\right],$$

By differentiating Equation (5), we shall obtain the velocity of the diffusion flow coming from the surface, in the form of:(8)$$\frac{dm}{dt}=-\beta m_{0}e^{-\beta t}.$$

The diffusion flow velocity changes into a time dependence, from the maximum at the initial instant, β·*m*_0_, to the zero one at complete vaporization. For the typical time *t_e_* = 1/β, the diffusion velocity is equal to β·*m*_0_/e or ~0.37·β·*m*_0_.

Below, we analyze the resultant equations by using parametric estimations with allowance for experimental data available from the literature sources.

## 3. Results and Discussion

### 3.1. Input Data for Estimations

The estimations used the data for hexogen (RDX), octogen (HMX), picramide (TNA), and trinitrotoluene (TNT) (Table 1) [14,19,20]. We employed the following as the input data for vaporization coefficient, film geometry, and initial temperature: α = 0.5, film thickness *h* = 100 nm, film surface area *S*_0_ = 1 cm^2^, and temperature *T* = 293 K.

It bears mentioning that different literature sources differ in data on the heat of vaporization. For instance, the latest study [18] determined the TNT heat of vaporization to be *H* = 494 kJ/kg, which is higher than that in other reports. Those authors explain it by a greater impact of temperature on the mobility of the surface molecules of TNT at the nanoscale, the molecules the present study dealt with.

### 3.2. Parametric Estimations

The effect of some parameters on the vaporization of RDX over time is evaluated below.

Consider the relationship between the mass of the explosive vapors and the time, as estimated by Equation (5), for three temperatures of the film (Figure 1). The vaporization at a normal temperature of 293 K is negligibly small (by the lapsed time of 30 min, ~10^−7^ m_0_).

The higher the film temperature, the greater the vapor mass (0.86 ng at 310 K and 18.23 ng at 340 K at 30 min).

Figure 2 depicts a relationship between the time over which the film has decreased by the *e* times, *t_e_*, and the temperature, for three values of parameter α.

The vaporization time is dependent non-linearly on the film temperature and parameter α. Parameter α determines the relation between the number of molecules escaping the surface and those coming back onto the surface. The smoother the surface, the easier the escape of the molecules from the surface under other equal conditions. This corresponds to a greater value of parameter α.

For detection methods, the threshold concentration of a substance is known. In the proposed model, we determine not the concentration, but the mass of the substance sufficient for detection, Δ*m_d_*. It is necessary to explain how the vapor mass Δ*m_d_* is related to the threshold concentration of the detection methods. The concentration of molecules in the air depends on the volume in which this mass is distributed. In various methods for detecting vapors of a substance, this volume can be different. In the case of sampling-based methods, this is the volume of the sampling chamber. Remote methods for detecting vapors (for example, laser-optical) determine the concentration of vapors concentrated near the surface in the diffusion boundary layer. The thickness of this layer, and hence the volume, depends on the aerodynamic conditions. The concentration in this layer will be Δ*m_d_*/*V_bl_*, where *V_bl_* is the volume of the boundary layer.

If the vapor mass sufficient for detection, Δ*m_d_*, is known, it is possible to identify the time within which such a mass of vapors is formed in the air, in line with Equation (6). Figure 3 displays the detection time *t_d_* plotted versus the ratio between the substrate surface area and the film surface area. The estimation was carried out for Δ*m_d_* = 0.1 ng, α_0_ = 0.9.

The more pronounced the surface porosity (a higher ratio of *S_s_*/*S*_0_), the more time is required for the specified amount of RDX vapors to appear in the air. If the surface is rather smooth and the air temperature is normal (293 K) it takes 1 h, to reach an amount of explosive vapors of 0.1 ng. However, this time may appear to be inappropriate when developing express systems for explosives detection.

### 3.3. Comparison of Sublimation Behaviors of Different Explosives

The sublimation behaviors of the layers (*S*_0_ = 1 cm^2^, *h* = 0.1 μm) of different explosives at *T* = 293 K and α = 0.5 are compared below (Table 2). They greatly differ in basic parameters influencing the sublimation rate, i.e., the heat of sublimation and the pre-exponential factor. TNA vaporizes more intensely, which is due to its lower heat of vaporization. The other two explosives are almost alike in the heat of vaporization, about 586–588 kJ/kg. That said, the pre-exponential factor of HMX is two orders of magnitude higher than that of RDX.

The tabulated data are consistent with the fact that HMX almost does not sublimate under normal conditions and the detection of its vapors under these conditions proves to be impossible. The other explosives, except for TNA, sublimate slowly under normal conditions.

The traces of explosives similar to RDX and TNT in characteristics (heat of sublimation, pre-exponential factor) are detectable by sensitive detectors even at room temperature. Low-volatile explosives such as HMX require finding detection methods that respond to the vapor mass of less than 10^−8^ ng.

The given estimations demonstrate that the degree of vaporization and the mass of the resulting vapors are dependent, considerably and non-linearly, on the parameters that are characteristic of explosive vapor pressure, film temperature, and substrate surface structure.

The results of the work can be used in assessing the concentration threashold of the developed explosive trace detectors. In methods based on air sampling, a threshold value for the concentration of a substance is known. Knowing it, one can calculate the minimum mass of the film, which can be determined under given conditions.

For methods of remote express detection, for example, based on the use of laser radiation, it is also important to know the time during which the amount of evaporated substance in the air will be sufficient to determine.

## 4. Conclusions

Here, we introduced a mathematical model for the vaporization of a thin explosive film with allowance for the substrate surface structure. Analytical equations were derived for the vapor mass change over time and for the vaporization time to attain the specified vapor mass sufficient for detection.

Calculations were carried out for the vaporization dynamics and typical length of vaporization plotted against the temperature and substrate surface area for a thin RDX layer with the area *S*_0_ = 1 cm^2^ and thickness *h* = 0.1 μm. When the film temperature is 293 K and the vapor mass required for detection is Δ*m_d_* < 0.1 ng, the sublimation time will be less than 70 s.

The typical time of sublimation and the amount of vapors in the air were estimated for four explosives (RDX, HMX, TNA, TNT) to show that the results were dependent considerably on the heat of vaporization. The thin layers of explosives with indicated properties similar to RDX and TNT can be detected even at room temperature. However, low-volatile explosives such as HMX give off a lesser mass of vapors. This means that such explosives necessitate finding detection techniques that respond to a vapor mass below 10^−8^ mg.

## Figures and Tables

**Figure 1 molecules-27-07939-f001:**
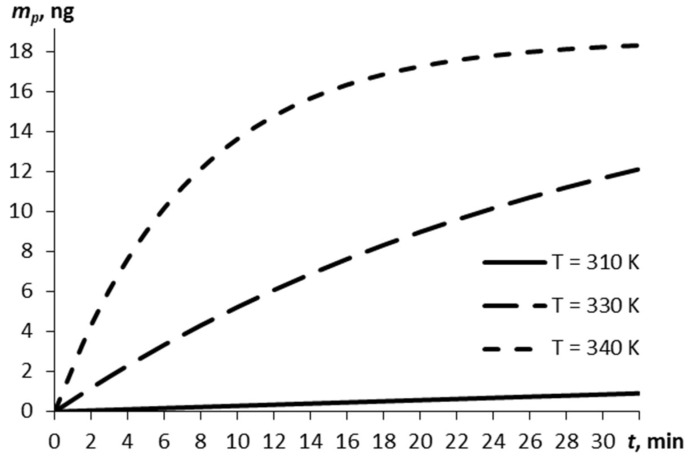
The vapor mass over time for different temperatures of the RDX film.

**Figure 2 molecules-27-07939-f002:**
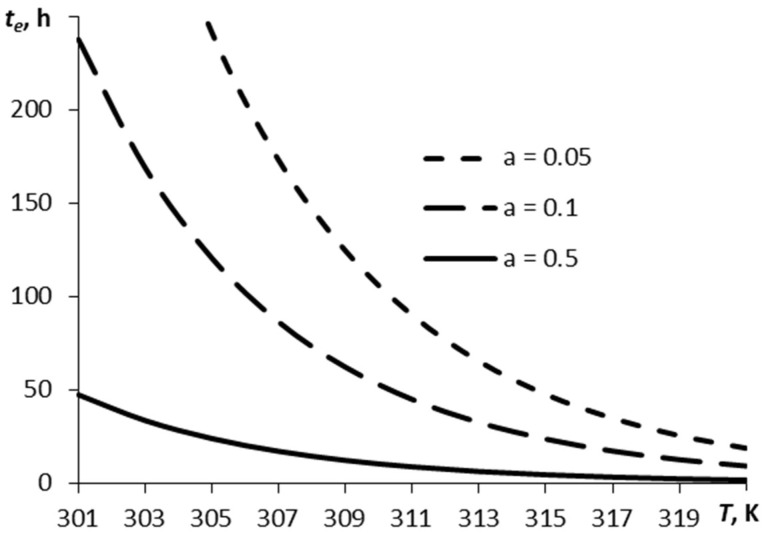
A typical time of sublimation plotted against temperature for different values of parameter α (RDX).

**Figure 3 molecules-27-07939-f003:**
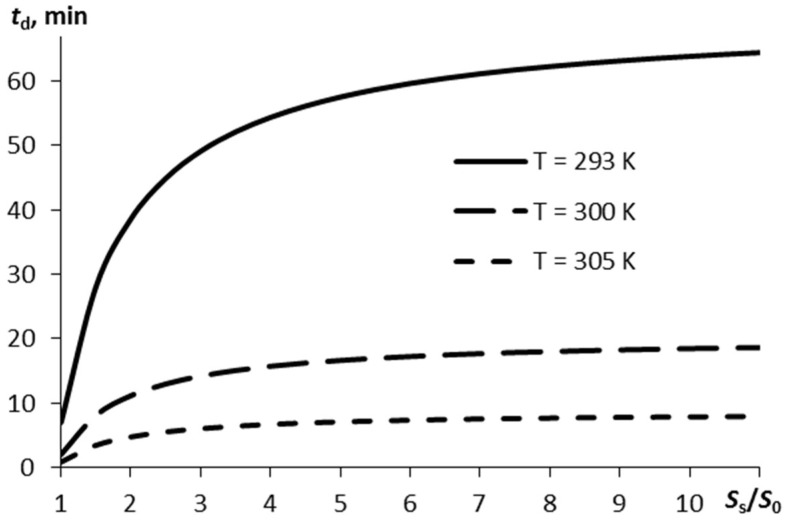
The time the vapor mass reaches Δ*m_d_* = 0.1 ng in the air on sublimation of the thin RDX layer as a function of the ratio of the substrate surface area to the explosive layer surface area for three different temperatures.

**Table 1 molecules-27-07939-t001:** Physicochemical properties of the explosives used.

Explosive	Vaporization Heat, kJ/kg	Pre- Exponential Factor *A*, Pa	Density ρ, kg/m^3^	Molecular Weight M, kg/mol	References
RDX	588	1.65·10^16^	1858	0.220	[19,20]
HMX	586	132·10^16^	1910	0.296	[19,20]
TNA	511	2.15·10^16^	1800	0.228	[20]
TNT	409	2.01·10^10^	1600	0.227	[14]

**Table 2 molecules-27-07939-t002:** Calculated sublimation behaviors of the explosives at *T* = 293 K.

Explosive	Vapor Mass in the Air after 30 min, ng	Typical Time of Layer Sublimation, *S*_0_ = 1 cm^2^ and *h* = 0.1 μm
RDX	0.05	195 h
HMX	6·10^−8^	10^8^ h
TNA	8.9	44 min
TNT	0.19	41 h

## Data Availability

Not applicable.

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
