# Peer review of "A Mathematical Model for Sublimation of a Thin Film in Trace Explosive Detection Problem"

_molecules, 2022, doi:10.3390/molecules27227939_

Round 1

Reviewer 1 Report

The paper by Olga Kudryashova and Sergey Titov proposes a mathematical model to evaluate the sublimation rate of thin films of energetic materials such as TNT, RDX, HMX, and picramide (TNA). This is important in view of the possibility of detecting explosives in traces in the gas surrounding a contaminated surface.

The paper uses the Hertz-Knudsen-Langmuir equation for the mass rate of vaporization. This equation describes the sticking of gas molecules on a surface by expressing the rate of change of the concentration of molecules on the surface as a function of the pressure of the gas. In principle this equation can be applied to thin film of explosives deposited on a substrates. A similar approach has been used by the same authors on the paper in reference [17]. It is not very clear which are the advancements of this paper with respect to the previously published paper (titles are almost identical). Moreover the paper is full of unclear statements and is not worth to be published in the present form.

I will list some of the drawbacks  found.

The Introduction  is too long. The authors underline the difference between the case of bulk explosive and thin films in which the interaction with substrate is relevant “To further advance the mathematical model, we suggest that this parameter be examined so that the substrate surface structure is factored in”. Nevertheless unclear explanations are proposed in the rest of the paper.

“The sublimation rate of an explosive, in particular TNT and RDX, was examined in several studies [12-17]”. The statement is correct but the references are related only to studies of explosive thin film sublimation, including a paper of the authors.

2.1 assumption: the author assume “It is important to note that the surface area of the film is much higher than the film thickness, S0 >> h”. How to compare a surface with a thickness?

“The pores require greater energy than the smooth surface for the removal of the explosive.”. This is not so trivial. One can also assume that a porous surface may improve the sublimation rate due to the increase effective area of the film. Is there any experimental confirmation?

2.2 Basic equation

Equation 1 is the Hertz-Knudsen-Langmuir equation, which expresses the film mass decrease  as a function of the pressure of the gas and other parameters due to sublimation.  The mass decrease rate depends on surface S, but as reported by authors the mass decrease affects the island size rather than the film thickness, as reported in ref [19], as if the mass decrease was due to the island edges rather that the island surface. The authors should clarify this point.

In equation 1 the coefficient α must be is the portion of the flow of molecules leaving  the surface rather than “the portion of the flow of molecules condensing on the surface”.

“The substrate surface area is linked to the porosity of the substrate material – specific surface area Sssa – via the relation: Ss = Sssa · ms, where ms is the mass of the substrate material”?? In which way may we correlate the surface with the mass of substrate material? Which is the physical dimension of parameter Ssssa?

“The parameter is inversely proportional to the root of absolute temperature, while the saturated vapor pressure ps decreases linearly with the temperature”?? Where can we understand this point?

“Hence, the velocity of diffusion flows, as measured in that study, is lower than those obtained in the other studies. Those authors explain it by a greater impact of temperature on the mobility of the surface molecules of TNT at the nanoscale, the molecules the present study dealt with”. Which is the meaning?

Finally the authors focuses their attention to the parameter” vapor mass sufficient for detection, Δmd,” while a more natural parameter to establish the detection possibility is the molecules concentration in air or the partial pressure of the explosive molecules.

“However, even if  the surface is rather smooth and the air temperature is normal (293 K) for about 1 h, the amount of the explosive vapors will turn out to be 0.1 ng. But this time may appear to be inappropriate when developing express systems for explosives detection. What does it mean= ??? Probably the authors want to say the time to evaporate 0.1 ng at T=293 °K is too long for an effective explosive detection.

Reference [18] is a dictionary?

In conclusion it seems that the paper is not suitable for publication on molecules

Author Response

Dear Reviewer, 

 The authors are really grateful to the respected reviewer for his attention to our work and valuable comments. They certainly allowed us to improve our manuscript. We hope that you will find our manuscript suitable for publication. Thank you for your attention.

Our response point by point is in the attached file.

 Best regards, 

Authors

Reviewer 2 Report

The manuscript is devoted to the development of a mathematical model based on the Hertz-Knudsen-Langmuir (HKL) equation for sublimation of explosive thin films. The author used 4 types of explosives when developing the model taking into account the influence of many factors: temperature, substrate porosity, adsorption parameters, etc. The obtained results have scientific novelty and practical significance. The following comments should be taken into account:

1. Significant improvement of the English is necessary.

2. Ln 93. As I understand it, the authors wanted to write that the films are so thin that their volume is not taken into account.

3. After equation (1), some parameters are not indicated.

4. Ln 140 – 142. Please indicate corresponding reference.

5. Ln 165. Indexes are incorrectly represented in the designation of quantities.

6. Is it probably worth presenting equations (4) and (5) for ΔS and Δm, respectively?

7. Is the density of the explosive a more important parameter than vaporization heat in determining the characteristic time? Table 1 and 2.

8. I propose to insert a column in Table 1 with corresponding references.

9. Ln 181. A typo in the numbering of formulas!

10. At the end of the manuscript, it is necessary to indicate which existing methods are applicable for detecting explosive vapors in the air at room temperature according to your results.

Author Response

(The authors gave the same response as above.)

Round 2

Reviewer 1 Report

The authors did a good job revisioning the papers.

The majority of comments and suggestions of the previous report are satisfactorily addressed.

Minor points are:

Page 1

“The explosive trace is the finest film of an explosive on the surface” finest or thinnest?

Page 3

“If the vapor pressure of the explosive is negligibly low in the ambient space, such an equilibrium does not arrive and the film will get vaporized completely.” The meaning is not clear.

Page 6

The higher the film temperature, the greater the vapor mass (0.86 ng at 310 K and 18.23 ng at 340 K at 30 min)

Page 7

“However, even if the surface is rather smooth and the air temperature is normal (293 K) for about 1 h, the amount of explosive vapors will turn out to be 0.1 ng”. Not clear meaning.

May be

“ if the surface is rather smooth and the air temperature is normal (293 K) it takes 1 h, to reach an amount of explosive vapors of 0.1 ng “

Author Response

Dear Reviewer, 

The authors are really grateful to the respected reviewer for his attention to our work and valuable comments. They certainly allowed us to improve our manuscript. We hope that you will find our manuscript suitable for publication. Thank you for your attention.

Best regards, 

Authors

  1. Page 1

“The explosive trace is the finest film of an explosive on the surface” finest or thinnest?

Yes, “thinnest”

  1. Page 3

“If the vapor pressure of the explosive is negligibly low in the ambient space, such an equilibrium does not arrive and the film will get vaporized completely.” The meaning is not clear.

We have tried to make this sentence clearer:

If the volume is not confined, the vapor molecules escape freely. Then the vapor pressure in the ambient is negligible. The sublimation rate exceeds the condensation rate and the film will get vaporized completely.

  1. Page 6

The higher the film temperature, the greater the vapor mass (0.86 ng at 310 K and 18.23 ng at 340 K at 30 min)

Corrected

  1. Page 7

“However, even if the surface is rather smooth and the air temperature is normal (293 K) for about 1 h, the amount of explosive vapors will turn out to be 0.1 ng”. Not clear meaning.

May be

“ if the surface is rather smooth and the air temperature is normal (293 K) it takes 1 h, to reach an amount of explosive vapors of 0.1 ng “

Yes thank you, that's much better, corrected.